# Indigenous knowledge of Rift Valley Fever among Somali nomadic pastoralists and its implications on public health delivery approaches in Ijara sub-County, North Eastern Kenya

**Geoffrey Otieno Muga**[1]*, **Washington Onyango-Ouma**[1], **Rosemary Sang**[2], **Hippolyte Affognon**[3]

**1** Institute of Anthropology, Gender and African Studies, University of Nairobi, Nairobi, Kenya, **2** International Centre for Insect Physiology and Ecology (ICIPE), Nairobi, Kenya, **3** CORAF, Dakar, Senegal

* Geoffreymuga@gmail.com

**Data Availability Statement:** All relevant data are within the manuscript.

## Abstract

Rift Valley Fever (RVF) is a zoonotic disease whose outbreak results in heavy economic and public health burdens. In East Africa, RVF is mainly experienced in arid and semi-arid areas predominantly inhabited by the pastoralists. These areas experience sudden, dramatic epidemics of the disease at intervals of approximately 10 years, associated with widespread flooding and the resultant swarms of mosquitoes. Pastoralists' indigenous knowledge and experience of RVF is critical for public health interventions targeting prevention and control of RVF.

The study adopted a descriptive cross-sectional design combining both quantitative and qualitative methods of data collection. A total of 204 respondents participated in questionnaire survey and 15 key informants and 4 focus group discussions were interviewed and conducted respectively. In addition, secondary data mainly journal publications, books, policy documents and research reports from conferences and government departments were reviewed. Findings indicated that the Somali pastoralists possess immense knowledge of RVF including signs and symptoms, risk factors, and risk pathways associated with RVF. Ninety eight percent (98%) of respondents identified signs and symptoms such as bloody nose, diarrhea, foul smell and discharge of blood from the orifices which are consistent with RVF. Heavy rains and floods (85%) and sudden emergence of mosquito swarms (91%) were also cited as the major RVF risk factors while mosquito bites (85%), drinking raw milk and blood (78%) and contact with animal fluids during mobility, slaughter and obstetric procedures (77%) were mentioned as the RVF entry risk pathways. Despite this immense knowledge, the study found that the pastoralists did not translate the knowledge into safer health practices because of the deep-seated socio-cultural practices associated with pastoralist production system and religious beliefs. On top of these practices, food preparation and consumption practices such as drinking raw blood and milk and animal ritual sacrifices

**Funding:** This research was funded by the International Development Research Centre (IDRC), Grant Number: 105509-038 (RS). The funders had no role in study design, data collection, and analysis, decision to publish or preparation of the manuscript.

**Competing interests:** The authors have declared that no competing interests exist.

continue to account for most of the mortality and morbidity cases experienced in humans and animals during RVF outbreaks.

This article concludes that pastoralists' indigenous knowledge on RVF has implications on public health delivery approaches. Since the pastoralists' knowledge on RVF was definitive, integrating the community into early warning systems through training on reporting mechanisms and empowering the nomads to use their mobile phone devices to report observable changes in their livestock and environment could prove very effective in providing information for timely mobilization of public health responses. Public health advocacy based on targeted and contextually appropriate health messaging and disseminated through popular communication channels in the community such as the religious leaders and local radio stations would also be needed to reverse the drivers of RVF occurrence in the study area.

## Author summary

Rift Valley Fever is a viral disease that affects both humans and animals. It is categorized as one of the re-emerging and neglected tropical diseases that mainly affects the poor and marginalized populations that lack access to health services and are readily ignored. Humans usually get RVF through bites from infected mosquitoes. Infections also occur when humans are exposed to the body fluids, or tissues of infected animals. Hence the risk of infection is greatest when slaughtering in the context of traditional sacrificial practices. This is the major reason outbreak of RVF is commonly associated with people whose livelihoods revolve around livestock rearing. In East Africa, RVF is mainly experienced in arid and semi-arid areas predominantly inhabited by the pastoralists. These areas experience epidemics of the disease at intervals of approximately 10 years associated with Elnino events. Understanding the knowledge base of the people in terms of RVF signs and symptoms and risk factors and pathways is important for the adoption of effective prevention and control measures. This study findings suggest that even though the Somali nomads are adept at recognizing RVF, this knowledge has not been translated into appropriate health practices due to the deep-seated socio-cultural practices. Hence, there is need for health authorities to mount locally appropriate public health advocacy campaigns, empower the livestock keepers to report observable changes in livestock and environment using their mobile phone devices and promote cross-disciplinary studies on RVF.

## Introduction

Rift Valley Fever (RVF) is a zoonotic disease that not only affects cattle, camels, sheep and goats but also people and wildlife [1]. According to WHO, RVF is a disease that mainly affects disadvantaged populations that have limitations of accessing health services [2]. Consequently, these populations suffer poor health status and destitution that exacerbate their disease load. Most infections in humans take the form of mild fever although a few cases degenerate to fatal hemorrhagic disease [3,4]. In East Africa, RVF is mainly noted in arid and semi-arid areas as sudden, dramatic epidemics of the disease at intervals of approximately 10 years, associated with widespread flooding and the resultant swarms of mosquitoes [5]. This pattern of outbreak was experienced in 1997/98 and 2006/07 when massive outbreaks of RVFV occurred in North

Eastern Kenya and other parts of East Africa, both associated with El Nino events [6,7]. Recently in 2018, the outbreak again occurred in North Eastern Kenya.

Out of about 1,400 species of infectious disease pathogens of humans, nearly 60% are derived from animal sources, hence the importance of recognizing the role of livestock, companion animals and wildlife in the interactions between animals and humans [8]. In addition, two thirds of emerging pathogens are of zoonotic origin. Only by recognizing the importance of these human-animal interactions can we fully understand how to control these infections. The virus that causes RVF—Rift Valley Fever Virus (RVFV) is a member of the *Phlebovirus* genus, one of the five genera in the family *Bunyaviridae* [1,8]. It is related to mosquito–borne outbreaks and erupts during torrential down pours. Infections in humans are caused by bites from mosquitoes that are infected with RVF Virus. Exposure to body discharges of infected animals can also lead to infection in humans. Primarily, RVF is transmitted by *Aedes* mosquitoes which are its vectors.

A study on Rift Valley Fever [1] reported that the Somalis used the term *San dik*, which means "bloody nose" to refer to Rift Valley Fever. The study also [1] showed that goats had the highest morbidity rate in the 2006/2007 outbreak while sheep were the most affected by RVF outbreak in terms of mortality, fatality and incidence. Coughing, bloody diarrhea, salivation, lachrymation, pruritus and fever were also mentioned by the herders as other symptoms of RVF [1].

Study findings have also indicated the role of socio-cultural practices in RVF occurrence where animal ritual sacrifices are cited as a major predisposition factor to RVFV infection [9]. The *haj* festivals and *Eid–al–Adha* among the Muslim nomadic pastoralists have been found to enhance vulnerability to RVFV [9]. These sacrifices are executed where there are multitudes of people, thereby exposing them to attack by RVF virus especially where the animal was sick at the time of slaughtering. Gender considerations have also featured prominently in RVF research by virtue of the fact that the differential roles of men and women in pastoralist communities may predispose them to diseases such as RVF in different ways. Studies have pointed out that men are triple likely to be infected by RVFV than women [4,7]. However, this higher infection rate observed among men, they posited, was as a result of physiological factors only [4] otherwise there was no gender difference in susceptibility to diseases. In another study, it was found that nomadic pastoral work is highly labour intensive [10]. The tasks are performed largely by labour drawn from the family. Children are socialized on their role of taking care of the small stocks. Later they are assigned adult responsibilities in livestock rearing. The nature of labour division among the pastoralists means that everybody may be infected by RVF, yet most of the RVF interventions such as public health education have disproportionately focused on adults.

Scholars [1,11] seem to agree that even though a few studies have been conducted on RVF in endemic-prone areas such as North Eastern Kenya, such studies are still limited. There is also paucity of knowledge on gender and other socio-cultural dimensions of RVF. Furthermore, the control measures that have been mounted in Kenya since the first major RVF outbreak in 1997/98 have largely been characterized by surveillance and compilation of reports for the health authorities [12] with little recommendations on strategies for incorporation of local people's indigenous knowledge on prevention and control measures This article stands out to attempt to bridge these gaps by assessing indigenous knowledge base of the Somali nomadic pastoralists and how it can be practically integrated into an early warning system and used to design other public health delivery approaches during future epizootics in North Eastern Kenya.

## Methods

### Ethics statement

The research team explained the objectives of the study to the participants and those who voluntarily expressed willingness to be interviewed were issued with consent forms for signing before participating in the study. The explanations included making the participants aware that the study would lead to publications in journals and disseminations in conferences. Key informants whose information was found important as supportive evidence to warrant direct quotation in the study findings were issued with consent forms which they filled in and signed to allow for publication. Research permit for this study was obtained from the National Commission of Science, Technology and Innovation (NACOSTI) and the Kenya Medical Research Institute (KEMRI).

### Study area and population

**Location.**    This research was carried out in Gediluun sub-location in Sangailu ward, Ijara sub-County, Garissa County in the North Eastern part of Kenya (Fig 1). The other sub-locations found in Sangailu ward are Handaro, Matarba, Sangailu and Wakabharey while Ijara, Hulungho and Masalani are the administrative wards that together with Sangailu form Ijara sub-County.

Ijara sub-County was one of the major hotspot areas in North Eastern Kenya during the last outbreak of RVF in 2006/07 and this informed the selection of Ijara sub-County as the study site. Ijara sub-County is also one of the seven sub-Counties that form Garissa County. The sub-County borders Fafi sub-County to the North, Lamu County to the South, Tana Delta to the South West, Tana River to the West and Republic of Somalia to the East. Ijara is also one of the four constituencies forming Garissa County. The others are Fafi, Dujis and Lagdera. The sub-County lies approximately $1^0$ 7' S and $2^0$ 3 S and longitude approximately $40^0$ 4 E and $41^0$ 32'E [13] and it covers an area of 10,000 $km^2$. The area is semi-arid located between Tana River and the boundary to Somalia.

**Population and livelihood.**    Garissa County where Ijara falls covers 44,175 km sq. with a population of 623,060 out of which about 93,000 live in Ijara sub-County. Approximately 11,474 people live in Sangailu ward alone where Gedeluun sub-location is situated [14]. Over 90 percent of the inhabitants of the sub-County are Somalis who are dependent on cattle keeping as their chief source of livelihood [11]. The remaining residents (about 10%) are non-Somalis who work in the area as traders and employees of Non-Governmental Organizations (NGOs) and Government Departments. A majority of the population live in the rural rangelands where nomadic herdsmen practice pastoralism although there are a few sedentary people living in the nearby small town centres.

When the government declared cattle trade and movement as illegal in 2006/7 during the last RVF outbreak, serious economic losses of close to US $ 10 million were reported [15]. Notable challenges to pastoralist production system today include diseases, dilapidated infrastructure, pests and environmental stresses. Cattle rearing therefore make the best economic use of the land due to the prevailing ecological conditions, low population density and availability of rangelands.

### Research design

The study design was cross-sectional and descriptive combining quantitative and qualitative methods of data collection. Field work was conducted in two phases; the first phase involved survey research for quantitative data. This obtained baseline and quantifiable information on

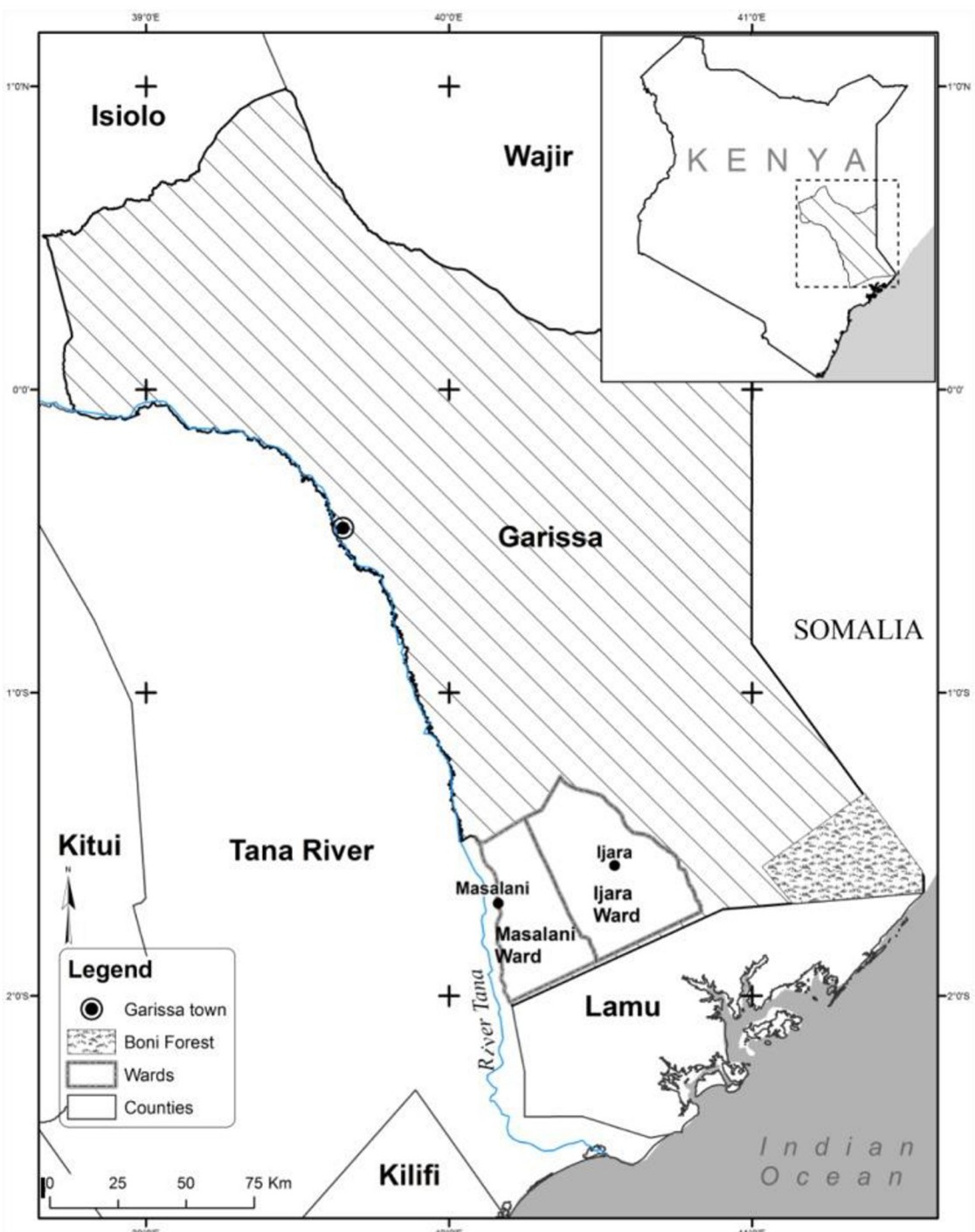

**Fig 1. Map showing Ijara sub-County study site.** Source: Abdi et al., (2016): Knowledge, Attitudes and Practices (KAP) on Rift Valley Fever among Pastoralist Communities of Ijara District, North Eastern Kenya. *Plos Neglected Tropical Diseases. 2015 Nov; 9(1 l):e0004239. DOI:* 10.1371/journal.pntd.0004239.

the demographics of the pastoralists. Pastoralists' production system, socio-cultural drivers of RVF and indigenous knowledge base on the zoonotic disease were also collected using survey method. The information collected from survey technique was used to sharpen and enrich the qualitative tools that were employed in the second phase of field work. This helped to yield in-

depth ethnographic and emic data on the interaction of ecology, livelihood and Rift Valley Fever disease. All the study instruments were piloted and pretested outside the study site before actual data collection.

**Study population and unit of analysis.** The study population consisted of herd owners in Ijara sub-County. The sample population was drawn from the study population and the unit of analysis was the individual herd owner.

**Sample population and sampling procedure.** The research team with the assistance of local administration identified a central point within Gediluun sub-location in Ijara sub-County. The team then took the coordinates of the central point (S $01^0$ 22.832' E $040^0$ 42.201 E 117) and administered survey questionnaires within 2 KM radius from the central point. The geographic reference system ensured efficiency in getting respondents who are mobile pastoralists since the sample frame for the sub-location was unknown. It also gave each potential interviewee within the radius an equal chance of being interviewed. Convenience sampling strategy was then employed to identify respondents within the demarcated radius. In terms of inclusion and exclusion criteria, those aged 18 and above (a majority age) and owning a herd were legible to participate in the survey study.

The mobility of the pastoralists was a major limitation in the study as it posed challenges of finding a large number of samples within the duration of field work. To address this, the precision/accuracy level was stretched from ± 5% (the standard) to ±7% and a manageable sample derived using Yamane (1967:886) formula as follows:

$$n = \frac{N}{1 - N(e^2)}$$

Where

$_n$ = Sample Size

$_N$ = Population size for the herd owners

$_e$ = Level of precision

$n = \frac{10000}{1 - 10000(0.07^2)} = 200 \text{ Herd Owners}$

Although the sample size according to the above formula is 200, the researcher managed to sample 204 herd owners for the survey.

## Methods of data collection

### Survey technique

A standardized questionnaire was administered on 204 herd owners aged 18 and above who were randomly sampled within a radius of 2 KM from the community central point within Gediluun sub-location. The researcher used local data collectors and community leaders to facilitate identification of the herd owners. The instrument was translated into Somali language for ease of interview and responses translated and recorded in English. The questionnaire had sections on thematic areas namely socio-demographics of respondents and knowledge on RVF signs and symptoms, risk factors, risk pathways, early warning events and incubation period. Other areas captured in the tool were food preparation and consumption practices, gender dimensions on RVF and implications for public health interventions.

### Key informant interviews (KIIs)

People who were knowledgeable in the livelihoods of pastoralists and Rift Valley Fever were the target of this method. An interview guide was used to collect data from 15 key informants

who were purposively sampled based on their expert knowledge. These interviewees comprised the human health and veterinary officials from government and NGOs, local administration officials, social development officers, district development officers and local authority officials. Others were herd owners and people who suffered from Rift Valley Fever outbreak in 2006/7.

## Narrative method

This method provided an account of people's experience with RVF. As such, people who suffered from the disease, had their livestock suffer from the disease or those who took care of RVF patients in the previous outbreak of 2006/7 were purposively sampled for interviews using this method. The key informants mainly the local administration and health officials were contacted for identification of narrative interviewees. The stories were told in the local Somali language and local interpreters were used to translate the information into English. A total of 5 narratives were collected by this method.

## Focus group discussions (FGDs)

This method was used to provide qualitative data to contextualize the study topic and build consensus on emerging issues from the other methods. FGD guide was used to facilitate the discussions. Forty eight (48) FGD participants who were herd owners and who did not participate in the survey were purposively sampled for discussions. A total of 4 FGDs (each comprising 12 participants) were conducted in the division (two with women and two with men). The local leaders and data collectors helped in identifying the FGD participants. Homogeneity of FGDs was observed by conducting women FGDs separately from those of men.

## Secondary data review

Secondary data sources mainly journal publications, books, policy documents and research reports from conferences and government's health and veterinary departments were consulted during the study. The review was undertaken to provide a background to the study, identify and frame the research problem and inform the design of study tools. Moreover, the review was instrumental in providing a basis for discussion of study findings.

## Data management and analysis

Data management was a continuous process during the study. Qualitative interviews were recorded using a voice recorder (with permission from the participants) and thereafter transcribed into word document with the support of a research assistant who was trained in the project. The transcriptions were reviewed by all the authors before cleaning and coding into running themes. To ensure more accuracy of information, the authors again reviewed and validated the coding and running themes that were generated at this stage. In terms of data analysis, qualitative data was analyzed by grouping the responses according to emerging themes mainly knowledge on; RVF signs and symptoms, risk factors, risk pathways, incubation period, food preparation and consumption practices and ritual sacrifices. Other thematic areas analyzed included knowledge on healing practices, cyclical mobility patterns, gender dimensions of RVF and implications for public health interventions. Verbatim quotes were pulled out from the data and outstanding ones that reveal the emic perspectives of the pastoralists on key study findings selected for use. Statistical Package for Social Science (SPSS) version 20 was used to analyze statistical data. Descriptive analysis method was employed to perform statistical analysis and this yielded descriptive statistics mainly percentages, averages and frequencies

of the measured variables. Data from different sources were triangulated to show the extent to which they related to one another in explaining the empirical world of the pastoralists.

## Strengths and limitations of the study

This study was conducted after the massive 2006/7 RVF outbreak in North Eastern Kenya. Although the findings supported most studies that have been conducted on RVF, it adds more vital evidence into RVF literature by consolidating the knowledge base of the pastoralists on RVF and its implications on public health delivery approaches. This contribution remains useful since the zoonotic disease continues to negatively impact on the health and livelihoods of the livestock keepers whenever it occurs after a cycle of every 10 years. Furthermore, this study stands out as it brings to the fore how the knowledge of the pastoralists can be utilized to inform public health delivery approaches. To prevent and control the spread of RVF during epidemics, the unique characteristics of the study population could greatly benefit from specific targeted health messaging using local radio stations, mobile phones and Muslim leaders which are popular communication channels in the community. The knowledge of the nomads can also be integrated into early warning system by training the nomads on a reporting system whenever RVF occurs and empower selected livestock keepers who own mobile phone devices to timely report to the health authorities any observable changes in their cattle and environment. In terms of limitations, due to convenience sampling strategy that was employed to respond to the mobility nature of the herd owners, the study population may not represent the true population of Ijara sub-County.

## Results and discussion

### Demographics of respondents

The study showed that pastoralist production system is closely intertwined with the cultural system of the pastoralists. Hence to better understand and contextualize the organization of pastoralist production system and the characteristics of study participants, a presentation of socio-demographic characteristics of herd owners is made in Table 1. The table shows that a majority of herd owners in the study area were men constituting close to 80% of the respondents. Women herd owners only accounted for 20% of those surveyed, a revelation that points at the patriarchal structure of the Somali society where male gender are culturally constructed as the resource owners. About 90% of the herd owners never attended school and this partly accounts for the pastoralists' beliefs and practices that influence the occurrence of Rift Valley Fever.

Slightly over half of the respondents (55.9%) were aged between 25 and 44 years old out of which 30% and 26% belonged to the age cohorts of 35–44 and 25–34 respectively. The two cohorts therefore comprised both the youthful and middle-aged herd owners. The elderly aged 45 and above constituted 40% of the herd owners. The youthful herd owners may have acquired the livestock through exchange for their labour for livestock as revealed in the quotation below:

*In this community, a majority of the youths are employed as mobile herders by the elderly and well off herd owners. In return for labour, one is given a bull after three months of moving with the herds.*

Source: A male key informant aged 24 and employed as a herder in Masalani village.

**Table 1. Socio-demographic characteristics of herd owners N = 204.**

| Variable | Category | n (%) |
|---|---|---|
| Sex | Male | 79.4 |
| | Female | 20.6 |
| Highest level of schooling | Never attended school | 89.2 |
| | Did not complete school | 7.4 |
| | Others | 3.4 |
| Religion | Muslim | 100 |
| Marital status | Single | 5.4 |
| | Married | 83.3 |
| | Widow/Widower | 7.4 |
| | Others | 3.9 |
| Average monthly income (Kshs) | Less than 5000 | 12.7 |
| | 5001–10,000 | 41.2 |
| | 10,001–15,000 | 29.4 |
| | 15,001–20,000 | 11.8 |
| | 20,001–50,000 | 4.4 |
| | No Answer | 0.5 |
| Ownership of communication gadgets | Mobile phones | 97.2 |
| | Radio | 48.6 |

Source: Survey Data

The young herd owners may have also been bequeathed with the animals from their older parents as is the practice in this patriarchal community. Because pastoralism is the main economic activity in Somali community, the youth who are employed as herders may start accumulating wealth at an early stage. About 83% of the respondents (herd owners) were also heads of households and of these, 79% were male. As male herd owners, they have the overall authority on the herd including health and when to sell any of the animals. Male herd owners are particularly responsible for treating the animals whenever they fall sick among other gender roles. This implies that targeting both herd owners and the youthful herders remains relevant for success of any public and veterinary health interventions especially during RVF outbreak. The role of moving with the herds exposes the youths to higher vulnerability to RVF during such cyclical mobility in search of water and pasture.

All the herd owners surveyed relied on pastoralism as their main source of livelihood. Average herd size was about 50 animals. Large herd production is considered both as a better coping mechanism and economic practice in the dry land ecosystem [12]. However, it is also a risk factor for RVF [16]. In terms of monthly income, close to one half (46%) of herd owners earned over Ksh 10,000 (US$100) per month from livestock rearing, a finding that underscores the pivotal role played by livestock in the economic lives of the pastoralists. The income includes proceeds from milk and cattle sales and other animal products. Income from animal production is supplemented by business which accounted for 77% of other sources of income for pastoralists who had diversified their livelihoods.

Study findings indicated that almost all herd owners (97.2%) owned and used mobile phones while close to half (48.6%) owned a radio. According to key informants, accessibility and ownership of these communication gadgets is very important in RVF control because the livestock keepers can easily exchange public health information with health authorities for timely response. Furthermore, they asserted that RVF reporting system and awareness creation

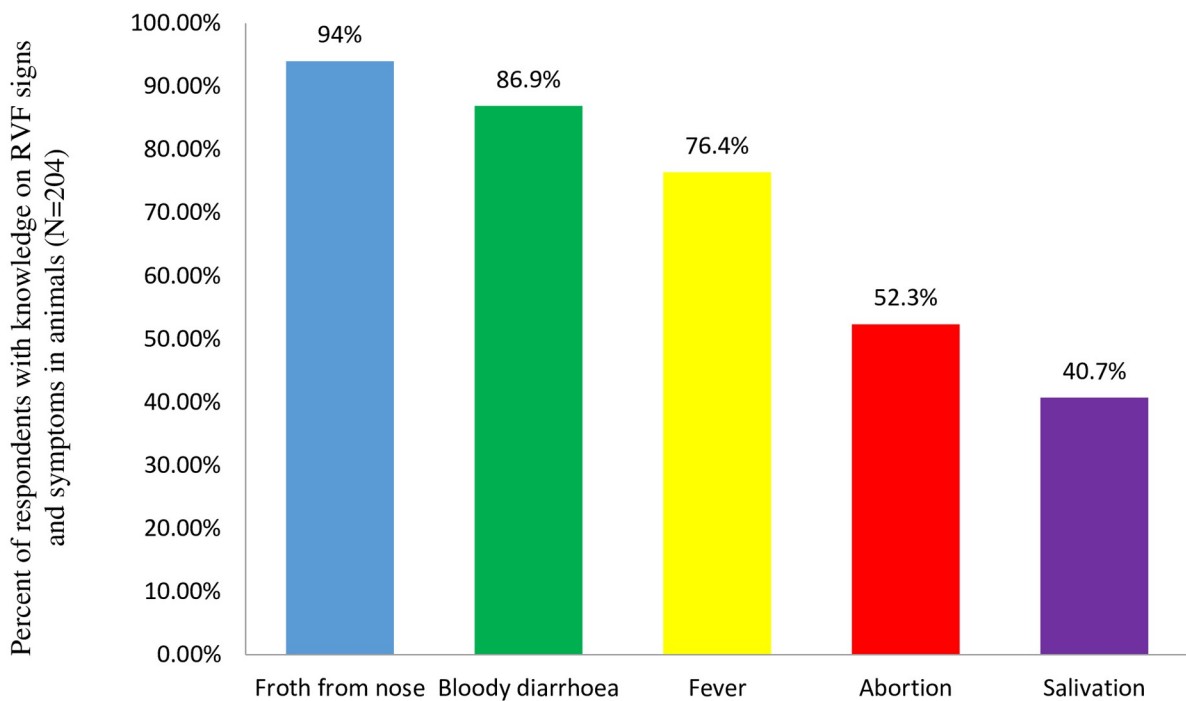

**Fig 2. RVF signs and symptoms in animals.**

can be immensely strengthened by the use of these gadgets. A study by [17] affirms this finding when it also indicated that 98% and 44% of the households in Ijara sub-County owned mobile phones and a radio respectively.

## People's knowledge base on Rift Valley Fever

**RVF signs and symptoms, risk factors and early warning events..**   Survey data revealed that 98% of respondents affirmed that they knew RVF and could positively identify its signs and symptoms in cattle, goats and sheep. These were froth from the nose (94%), bloody diarrhea (86.9%), fever (76.4%), abortion (52.3%) and salivation (40.7%). The signs and symptoms of RVF as expressed by the respondents are shown in Fig 2.

The risks factors and early warning events associated with RVF were also revealed by the respondents as emergence of mosquito swarms (84.8%), sudden torrential rainfall (91.2%), stagnant water (29.4%), biting flies (35.3%) and contact with wild animals (10.8%). These are presented in Fig 3.

A key informant interview with a community health worker further supported these survey findings:

*"In 2006/7 RVF outbreak, there were torrential rains with floods. Water was everywhere and roads were impassable. Then one to two weeks after the floods, swarms of mosquitoes emerged. I remember also beginning the second week to the third week, animals particularly sheep started becoming sick. In the fourth and fifth week, animals started dying followed by humans".*

Source: A male Community Animal Health Worker in Gedeluum Village.

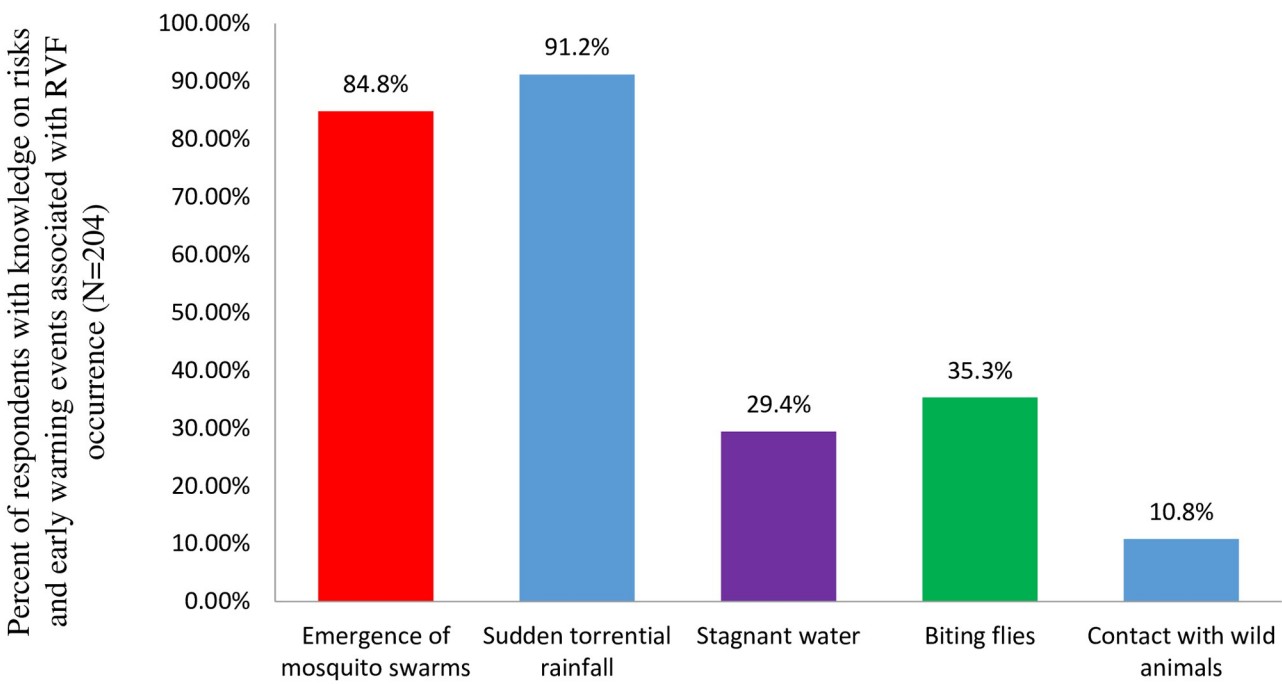

**Fig 3. Risk factors and early warning events associated with RVF occurrence.**

The data from both survey and key informant interviews demonstrates that the community has valid local knowledge on RVF and can positively discern RVF signs, symptoms and early warning events. These findings have implications for public health delivery approaches as the knowledge possessed by the nomadic pastoralists can effectively support early warning system established by the government for timely prevention and control of RVF outbreak. The early warning system that integrates the knowledge of the nomads can be operationalized if the government builds the capacity of the herd owners on reporting mechanisms in times of disease occurrence and improves its communication linkages with selected pastoralists who own mobile phones to immediately report any observable signs, symptoms and risk factors. The reports can then be validated by the authorities and made an integral part of a decision-making support tool to inform timely interventions such as sensitizing the livestock keepers on impending outbreak for timely adoption of preventive, coping and risk aversion mechanisms.

The above findings on RVF signs and risk factors are consistent with those of other scholars [1] who found that the cattle keepers had valid knowledge on symptoms of RVF and the associated risk factors mainly massive rains and mosquito swarms. They went on to add that the local people used the term *san dik* to refer to "bloody nose", which was a condition associated with the zoonotic disease. The people recollected and acknowledged that RVF was dangerous and had struck the area before in 1997/98. The cattle keepers expressed that a majority of their livestock were again sick in the one-year period of 2006/7 outbreak. This finding confirmed that of another study which reported that cattle keepers in Ijara sub-County recognized that the disease (RVF) is dangerous (99%), and had a positive attitude towards vaccination of animals (77%) [17]. Similarly, study findings have found out that availability of vectors, large number of cattle, and high rainfall were rated as most important and /or important risk factors associated with RVF [16]. A strong association between severe infections of RVF and handling of a large number of animals, closeness to water points and mosquitoes was also established in 2006/7 outbreak [18].

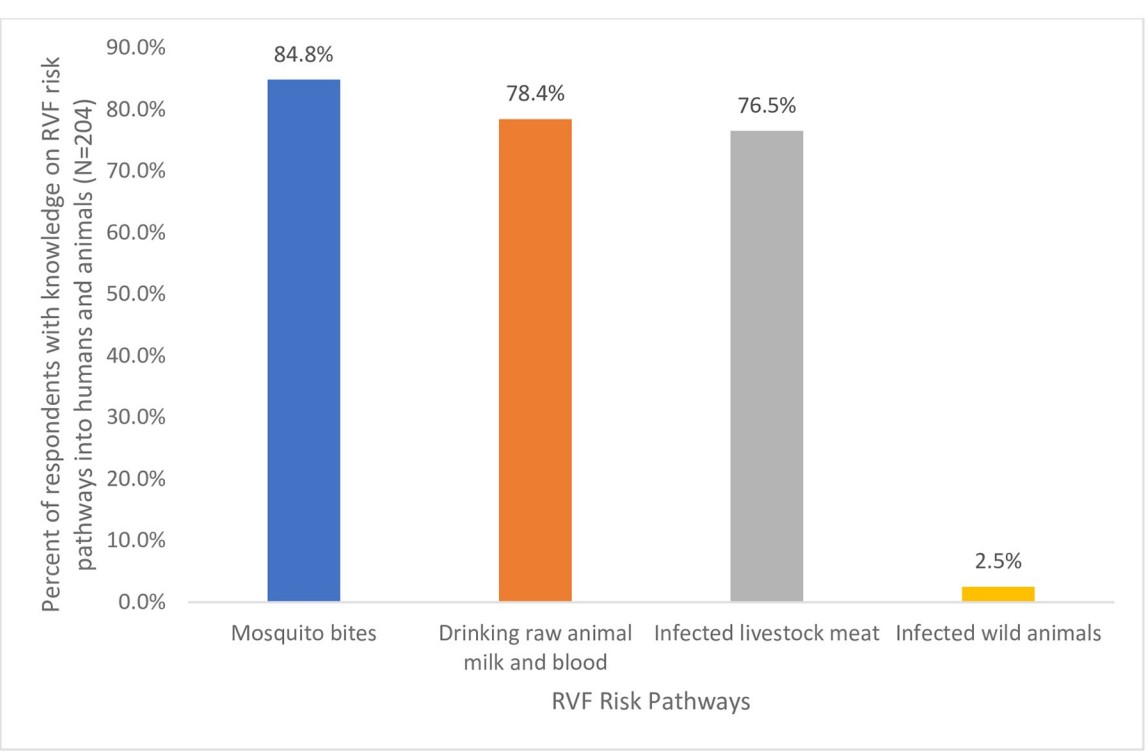

**Fig 4. RVF risk pathways into humans and animals.**

These study findings reveal the contribution that indigenous knowledge possessed by the lay people can make in public health surveillance. However, this contribution was not considered in the 2006/7 epidemic in Kenya but the authorities can benefit a great deal if it can be integrated into the early warning system since the cattle keepers had noted the changes in their environment and livestock prior to detection by public health surveillance systems [19]. This finding therefore strongly suggests that public health monitoring systems such as early warning systems could be immensely boosted to timely detect disease emergence by integrating the local knowledge of the local people. This will ensure that the authorities have sufficient lead time for mobilization of response before the cases escalate to full-blown outbreak.

**RVF *risk pathways and food preparations and consumption practices*.** The respondents were found to possess knowledge on RVF entry risk pathways as shown in Fig 4. Respondents mentioned mosquito bites (84.8%), drinking of raw animal milk and blood (78.4%), infected livestock meat (76.5%) and infected wild animals (2.5%) as the main risk pathways.

Indigenous knowledge pertaining to food preparations and consumption practices was found to play a key role in amplifying RVF infection and spread in the community. Salient was the practice of drinking raw blood and milk which they expressed are rich in nutrients as opposed to boiled ones. This finding seems to contradict the pastoralists' knowledge on RVF entry risk pathways. This contradiction may however, be attributed to the deep-seated socio-cultural beliefs and perceptions around raw milk and blood and lack of diversity food in sources. Raw milk and blood is perceived by the cattle keepers to be rich in proteins and vitamins that give the herders the required energy to move with the herd over long distances. Similarly as reported in another section of this work (healing prayers), the nomads believe that RVF is a curse and therefore those who eat infected cattle meat cannot die from the disease. These perceptions and beliefs have public health implication as they may predispose the

herders to RVF should the animals be infected with RVFV. Therefore, human and veterinary health officials may work with the pastoralists to identify the underlying reasons that support these perceptions and beliefs and develop relevant health messages to invoke appropriate behavior transformation. Demographic data has shown that 90% of the study population never attended school. This finding coupled with the fact that the nomads occupy difficult-to-reach and isolated settlements implies that such messages should be developed and disseminated in a local language for proper targeting and contextual appropriateness. Moreover, public awareness campaigns should involve the use of popular communication channels in the community mainly religious leaders and local radio stations for ease of uptake and impact.

These findings support those of another study in the same region which reported that the perceived entry risk pathways for RVF according to key informants were infected mosquitoes, infected domestic animals, infected aborted fetuses and fluids and infected wild animals [16]. It is important to note that RVFV is endemic in Ijara sub-County. As a result, the likelihood of its entry into Ijara through other pathways may not be applicable since it is already present in the mosquitoes [16].

The fact that RVFV is resident in the mosquitoes explains why the communities' perception about the source and /or spread of RVF is the same, namely the role played by mosquitoes.

The findings demonstrate that cattle keepers in Ijara are knowledgeable in RVF risk factors and risk pathways although some researchers' findings contradict these in their study [20]. They found that the nomadic pastoralists believed that infections in humans occurred as a result of mosquito bites and had little to do with their consumption of meat, milk and blood from infected livestock [20]. The participants in their study indicated that they had heard of the risks of acquiring the disease through consumption of livestock products but their experiences did not tally with the information they had received hence to them, Rift Valley fever was not transmissible through their dietary practices.

The findings on food preparations and consumption practices were affirmed by other scholars [16] whose respondents also mentioned the entry pathways for RVF as infected mosquitoes and infected domestic animals [16]. Therefore, infected animal fluids and products mainly raw milk and blood remain active conduits for RVF transmission from animals to humans and these call for the need for public health education using context specific advocacy tools, health messages and approaches.

**The effects of livestock sacrifice rituals on RVF occurrence.** Study findings showed that 76.5% of the respondents knew that eating infected meat is a major pathway to RVF infection. However, this important health knowledge seems to be lost in the more dominant livestock ritual sacrifices that characterize the daily lives of the Somali nomadic pastoralists and is therefore not translated into safer health practices. The Muslim pastoralists have a close symbiotic relationship with their livestock. As such livestock is central in their lives and plays a big role in ritual celebrations. The following quote underscores this finding:

*We Muslims slaughter animals to celebrate special occasions. The main one is Eid Al Adha. This is celebrated during the pilgrimage when the people access the holy place in Mecca. People back here in Ijara celebrate by slaughtering animals and giving the meat out to all the poor. Everybody slaughters as many animals as possible. Another celebration is the Ramadhan-Eid Al Fitr which follows 30 days of fasting. This is celebrated at family level and each family slaughters at least one animal.*

Source: A male key informant from Sangailu who is also a *Kadhi* in Sangailu Mosque.

It can be deduced from above quotation that livestock ritual sacrifices dominate Somali religious life. They are also performed when a religious leader is called upon to recite the Quran to the sick for healing purposes. Study participants reported that in the last RVF outbreak of 2006/7, religious leaders were invited by the affected families to recite the Quran to RVF victims so as to cure them. Other social occasions where animals are ritually slaughtered include weddings and during calamities like severe droughts. The respondents knew that eating infected meat could lead to RVF infection but this knowledge was defied due to the many animal sacrifices held in the community almost on a weekly basis. Affirming this finding, another study similarly indicated that contact with the sick livestock can happen when slaughtering or treating the animals hence the animal sacrifices among the Somalis present a health hazard [9]. These findings therefore explain the reason why RVF outbreak is commonly associated with the nomadic pastoralists whose livelihoods primarily depend on livestock. They also reinforce the fact that there is a close interaction of ecology, livelihood and the zoonotic disease (RVF). Even though livestock ritual sacrifices perform religious and social roles in the Somali community, in terms of public health they could act as drivers of vulnerability to RVF and risk pathways to human infection particularly if the animal slaughtered is infected with RVF Virus.

**RVF healing practices.** The study further indicated that some of the nomads believe that RVF is a curse and whoever becomes a victim requires the intervention of the Supreme Being *(Allah)*. The people said that the intervention by *Allah* can only be secured by a spiritual leader *(Sheikh)* reciting the Quran in a ritual prayer to the victim after which an animal is slaughtered to cleanse the victim. The following quotation by a local male *Sheikh* further underscores this fact:

> *Some people here believe that RVF is a curse. For this reason and despite public health education, they continued to eat uninspected meat during the last RVF outbreak arguing that those who died of the disease had been cursed. They also said that it is Allah who gives life and knows when somebody would die. Some of the people here even said that they also ate the same meat with those who died but they were not infected and are alive to date.*

Source: A male key informant who is a *Sheikh* in Matarba Village.

From the data above, it is apparent that the belief in the causation of RVF and *Allah* as the provider of life may have far-reaching health implications. As reported by the key informant, the belief that RVF is a curse that can be cured through a ritual prayer was the main reason that led the study population to defy public health information during 2006/7 RVF outbreak. The ritual prayers can be understood and analyzed within the framework of health system model that points out that people seek health within a local cultural system comprising three overlapping parts: the professional, folk, and popular sectors [21]. In this case, the ritual prayers fit within the folk sector. Even so, they could lead to delay in accessing conventional medical care and adoption of safer health practices. It can therefore be argued from the findings that some of the indigenous knowledge, beliefs and practices of the Somali nomadic pastoralists are not consistent with conventional medical knowledge and may compromise the uptake of preventive and curative measures within the realm of conventional medicine. This revelation needs interventions through public health awareness and health messaging to address the knowledge gaps and associated practices.

**RVF incubation period and severity.** Study data from the first quotation by the Community Animal Health Worker (above) shows that the local people had a valid knowledge on Rift Valley Fever risk factors and events preceding the outbreak including its incubation period. From their recall of 2006/7 RVF outbreak, they provided information on the periods marking

progression of RVF occurrence from the onset of floods through emergence of mosquito swarms, the first morbidity in sheep to full-blown outbreak in animals and humans. Furthermore, close to 73% of the respondents did not have a history of RVF in their households and hence gave information based on what they observed and experienced in the community. Only 26% reported that they and/or their family members had suffered from RVF. 52.8% of those who had a history of RVF in their households additionally said that they suffered from the zoonosis about 10 years ago. This period almost coincides with the 2006/7 RVF outbreak in N.E. Kenya thereby confirming clarity in recall and validity of knowledge from those respondents who had cases in their households.

The above findings are consistent with those of a study on the 2006/7 RVF outbreak in N.E. Kenya that noted that the livestock keepers gave accurate time lines of 2006/7 RVF events development that could have proved useful to the authorities to mount early response [1]. The livestock keepers were also able to vividly recall duration of RVF evolution in their locality. They recalled that the duration taken from the onset of massive rains to the initial appearance of mosquito swarms was about three weeks whereas the duration taken from the emergence of mosquito swarms to the first detected case of RVF in cattle was about two weeks. These intervals based on community knowledge reveal sufficient lead time which can effectively be used by the authorities to mount early response. Hence documentation of peoples' experiences and the lessons learnt in 2006/7 remains critical for the authorities to use in informing future interventions in terms of surveillance, establishment of decision-making support tool and management of future outbreaks.

Study findings showed that the top four severe livestock diseases in Ijara sub-County as mentioned by the respondents are Contagious Caprine Pleuropneumonia *(hawadeeda)* (81.9%), Foot and Mouth Disease (*Cacbeeb or habeeb*) (80.9%), RVF (*Sandik*) (74%) and Trypanosomiasis (*bukaanka jiifka)* (56.9%), a fact that underscores that the people know that RVF is one of the top four diseases that is perceived as severe and a big threat to their livelihood and health. Key informants and FGD participants also said that the community view RVF as "dangerous" hence the name *san dik* meaning "a bloody nose". Blood is a sign of danger and hence RVF is considered dangerous in the community. The severity that the nomads attach to the disease reveals a positive attitude that also underscores the importance of mounting preventive and control measures against RVF.

## Gender role as a predisposition factor to RVF

Eighty four (84%) of the respondents knew that herders, who are mainly young and youthful men tasked with the responsibility of moving with cattle in search of pasture and water were the most vulnerable to RVF during an outbreak. When the respondents were asked why they thought that the herders were the most vulnerable to RVF, they gave the following reasons: the herders interact closely and regularly with animals as they look for pasture and water, they get exposed to animal fluids when slaughtering animals, they handle lactating animals while out in the rangelands, and they together with their herds interact with wild animals in the forest during their cyclical movement thereby multiplying transmission from the wild. These findings therefore suggest that gender roles, among other factors, are closely associated with vulnerability to RVF. Table 2 presents findings on how gender roles are constructed for men, women and children among the Somali community.

Field data therefore showed that the roles of men predispose them more to RVF infection than women although in another study, it was deduced that the high infection rate among men may also be traceable to physiological factors [4]. The findings of this study support those

**Table 2. Gender role matrix in pastoralist production.**

| Gender Roles | % Response for Men | % Response for Women | % Response for Children |
|---|---|---|---|
| Moving with animals in search of pastures and water | 33.2 | 0.6 | 26.6 |
| Treating livestock when sick | 31.4 | 1.5 | 1.1 |
| Providing security for the animals | 27.0 | 2.1 | 1.7 |
| Milking of cattle | 0.8 | 60.8 | 39.3 |
| Others | 7.2 | 34.7 | 28.1 |
| No Answer | 0.2 | - | 3.2 |
| Do not Know | 0.2 | 0.3 | - |
| Total | 100 | 100 | 100 |

of many other scholars [4,7,22,23] who have attempted to unravel the grey area between disease occurrence and the sex of the victim while also contradicting the results of others [10,24].

For example, some studies have pointed out that women assume management responsibilities for the livestock while men because of patriarchal reasons assume the role of the owner of the animals and therefore wealth [24].

These differential social positions occupied by both gender mean that women may be more exposed to RVFV since their daily chores of milking among others bring them closer to the animals. These responsibilities (milking and processing milk and hides and skins) revolve around handling animal tissues and fluids and may differentially predispose them to RVF more than men. Despite the contradictions, many studies seem to affirm the finding on the disproportionate vulnerability of men to RVF compared to women [7,22,23]. Men have been found to be triple likely to get infection than women, a situation also observed in Kenya during the 1997 RVF epidemic [7]. Similarly, it has been reported that nomads particularly men have relatively higher chances of contracting Neglected Tropical Diseases (NTDs) compared to sedentary people [22] even though the mobility may also make them evade some diseases [23].

Drawing from the findings of this study, vulnerability of men particularly the youthful male herders could be understood within the wider context of their gender role and food preparation and consumption practices. The role of the herders puts them in close proximity to livestock when they are traversing the rangelands. While out in the fields, the herders depend on the cattle products mainly raw milk and blood which present a health hazard as they increase their risks of contracting RVF than women and herd owners who remain in the homesteads. This important finding has implication on prevention and control measures to reduce morbidity and mortality on both humans and animals during future epizootics. Health awareness campaigns and treatment against RVF should not only target women and the elderly herd owners who are found within the settlements but also the mobile youthful herders in the expansive rangelands who as the study has shown are more vulnerable to RVF exposure due to their gender roles. Moreover, the seemingly differing findings and arguments on the connection between disease outbreak and gender of the infected persons could be pointing at the need to undertake more studies in this area to generate evidence to inform public health programmes and policy.

## Conclusions

This article has assessed indigenous knowledge base of the Somali pastoralists and its implications on public health delivery approaches. Even though the findings of this study support those of other researches undertaken on RVF, they have made a unique contribution by detailing how the knowledge base of the pastoralists can be used to inform public health delivery

approaches so as to reduce the impact of future epizootic. The study has found out that the Somali pastoralists were adept at recognizing RVF signs, symptoms, risk factors and risk pathways. It has also shown that the pastoralists were knowledgeable on the role played by their socio-cultural practices such as drinking raw blood and milk in driving RVF occurrence. Furthermore, the dominant practices around animal ritual sacrifices, ritual prayers and religious beliefs were found to have influence on RVF occurrence. The study also indicated that the community was aware that men particularly the youthful herders are more susceptible to RVF than women and the elderly herd owners. However, the study has strongly indicated that this important body of knowledge has not been used by the people to transform their health behavioral practices. Finally, survey findings have indicated that almost all the herd owners possess and use mobile phones while close to half (50%) own radio. These communication resources have the potential to strengthen RVF control and management measures.

The above findings have significant implications for public health interventions. The findings on RVF signs and symptoms and risks factors imply that the community may play a big role in supporting early warning system by reporting the signs and symptoms and entomological, climatic and vegetation changes to the authorities. This has the potential to allow the authorities sufficient lead time for mobilization of response. To operationalize an early warning system that integrates the knowledge of the nomads, the government should train the herd owners on reporting mechanisms in times of disease occurrence and improve its communication linkages with the people by empowering a selected number of them who own mobile phone devices to immediately report any observable changes in livestock and environment to the health authorities. The reports can then be validated by the authorities and made an integral part of a decision-making support tool to inform timely interventions such as sensitizing the livestock keepers on impending outbreak to enable timely adoption of preventive, coping and risk aversion mechanisms.

The study has shown that the pastoralists are not only adept at identifying RVF risk pathways but they also know that their socio-cultural practices such as drinking raw milk and blood can amplify RVF transmission and spread. However, this important body of knowledge has not been translated into safer health practices mainly because of the deep-seated religious and socio-cultural beliefs. Therefore, public health campaigns that utilize clear health messages to challenge the predisposing practices to RVF are required to realize better health outcomes for the study population. Human and veterinary health officials may work with the pastoralists to identify the underlying reasons that support the perceptions and beliefs in the food practices and RVF causation and develop relevant health messages to invoke their abandonment and uptake of safer dispositions. Such messages should be well targeted and made contextually appropriate to reach a majority of the local people who are illiterate and the vulnerable and mobile youthful herders. They should also be developed and disseminated in a local language. Moreover, effective public awareness campaign strategies should involve the use of Muslim leaders and local radio stations that are popular communication delivery channels in the community.

Finally, this study makes a unique contribution in its efforts to bring a multi-disciplinary approach to the study of RVF. The authors comprise of anthropologists, a virologist and an economist whose orientations have been brought to bear in the study. As such the study adds more vital and multi-faceted evidence into RVF literature by consolidating the knowledge base of the pastoralists on RVF and its implications on public health delivery approaches. This contribution remains useful since the zoonotic disease continues to negatively impact on the health and livelihoods of the livestock keepers whenever it occurs after a cycle of every 10 years. This makes a multi-disciplinary approach to the study of RVF more relevant, sustainable and effective than never before.

## Acknowledgments

We recognize the support received from the International Centre for Insect Physiology and Ecology (ICIPE) in facilitating the authors' access to research institutions in Kenya for some of the literature referenced in this work. We are also sincerely grateful to the community of Ijara sub-County and staff of government departments who gave us valuable information during field data collection phase. Finally, many thanks to the ICIPE Eco-Health project team including James Wauna for giving field support and technical ideas which shaped the manuscript.

## Author Contributions

**Conceptualization:** Geoffrey Otieno Muga, Washington Onyango-Ouma.

**Data curation:** Geoffrey Otieno Muga.

**Formal analysis:** Geoffrey Otieno Muga.

**Funding acquisition:** Rosemary Sang.

**Investigation:** Geoffrey Otieno Muga.

**Methodology:** Geoffrey Otieno Muga.

**Project administration:** Rosemary Sang.

**Resources:** Rosemary Sang.

**Supervision:** Washington Onyango-Ouma.

**Validation:** Rosemary Sang.

**Writing – original draft:** Geoffrey Otieno Muga.

**Writing – review & editing:** Geoffrey Otieno Muga, Washington Onyango-Ouma, Hippolyte Affognon.

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
