## [Decision Letter · Decision Letter 0]

15 Apr 2020

Dear Dr Muga,

Thank you very much for submitting your manuscript "Indigenous knowledge of Rift Valley Fever among Somali nomadic pastoralists and its implications on public health delivery approaches in Ijara Division, North Eastern Kenya" for consideration at PLOS Neglected Tropical Diseases. As with all papers reviewed by the journal, your manuscript was reviewed by members of the editorial board and by several independent reviewers. In light of the reviews (below this email), we would like to invite the resubmission of a significantly-revised version that takes into account the reviewers' comments. 

I leave the decision to modify the title and structure of the abstract to the authors. Editorially, I think it is fine to leave it as it.

We cannot make any decision about publication until we have seen the revised manuscript and your response to the reviewers' comments. Your revised manuscript is also likely to be sent to reviewers for further evaluation.

Sincerely,

Benjamin Althouse

Deputy Editor

Scott Weaver

Deputy Editor

I leave the decision to modify the title and structure of the abstract to the authors. Editorially, I think it is fine to leave it as it.

Reviewer's Responses to Questions

**Key Review Criteria Required for Acceptance?**

**Methods**

-Are the objectives of the study clearly articulated with a clear testable hypothesis stated?

-Is the study design appropriate to address the stated objectives?

-Is the population clearly described and appropriate for the hypothesis being tested?

-Is the sample size sufficient to ensure adequate power to address the hypothesis being tested?

-Were correct statistical analysis used to support conclusions?

-Are there concerns about ethical or regulatory requirements being met?

Reviewer #1: (No Response)

Reviewer #2: The objectives of the study were clearly stated in the author summary and introduction of the manuscript. The authors state that studies on the knowledge and practices of Rift Valley fever in nomadic pastoralists are still lacking, especially the Somali pastoralists of NE Kenya. The authors also state that by incorporating the pastoralist knowledge into an “early warning” surveillance system then valuable time may be gained to implement prevention measures from Rift valley fever, therefore limiting disease spread and focus of control measures.

The study design seems to be based on standard observational and focus group and interview research methods. It seemed appropriate for the type of study and expected data and results the authors expected to obtain. The sample size calculations and statistical methods used also seemed appropriate. 

The results were presented primarily in narrative form and key quotations or themed observations were presented. The data presented in tables and graphs was appropriate for the study conducted. 

No ethical concerns for how the study was carried out or how the data and observations were presented in the paper. Proper IRB approval was obtained and informed signed consent was obtained form the study respondents.

Reviewer #3: The objective of this report was clearly stated as studying the RVF knowledge base of the Somali nomadic pastoralists in the Ijara district, using both quantitative and qualitative methods. They clearly defined how they recruited participants, but did not provide demographics- specifically of the 204 herd owners who completed the survey. This information should be displayed in a table containing variables such as age, education, gender, work roles (herding only, slaughtering, birthing, etc..) to better understand the study population. It is also unclear what their inclusion and exclusion criteria were. The authors state that they “mapped out enumeration area”, but they do not state how many herd owners were in the defined 2k radius or if enumeration actually occurred. However, they state that “convenience sampling strategy” was used- which is more accurate. The authors then go on to describe a detailed sample size calculation which was not met (385), then employed a different technique to justify a sample size of 200. Given that the results reported from their survey are purely descriptive in nature and without statistical analysis, would consider just describing their sampling methods and describing recruitment of a migratory population as a limitation in the discussion. They could also mention how many participants were approached but did not participate. 

The description of secondary data review was vague and unclear how this data was actually used. Was it used to develop the survey or to guide the interviews?

The data management section could be more concise regarding qualitative methods. It is not important for the reader to know that data was stored on flash disks and compact disks. This section should elaborate more on how the interviews conducted, reviewed and coded to identify themes. What topics were discussed and how were the interviews and focus groups guided? How were direct quotes selected? Were multiple reviewers/coders involved?

**Results**

-Does the analysis presented match the analysis plan?

-Are the results clearly and completely presented?

-Are the figures (Tables, Images) of sufficient quality for clarity?

Reviewer #1: (No Response)

Reviewer #2: The analysis of the focus groups and structured interviews was straightforward as it was presented and analyzed. Only a few comments. 

- The results were written in a descriptive format and focused on the observations and calculations of the study participants knowledge and perceptions of the risk of RVF. Although this was appropriate for this study it lacked detailed analysis to be able to draw significant conclusions for validating their findings

- A majority of the studies findings essentially validate or duplicate findings and knowledge of Rift Valley fever that have been known and observed for a very long time. Overall, the findings and results are not “new” or elucidate any new observations for the risk factors or knowledge of prevention methods for RVF

- The results are clearly stated and correspond to the study objectives and relay the studies intent

- The figures also relay the studies findings clearly. Although, they are simple bar graphs and do not significantly add much value to the visualization or understanding of the findings. 

- Inclusion of a map of the study area would have been helpful to understand the study location in relation to Somalia and past RVF outbreaks that were referred to in the analysis and discussion.

Reviewer #3: Overall, the results and discussion section should be separated into separate sections. There should be a section and table showing survey participant characteristics, especially because they are presenting descriptive statistical findings. The separate figures presenting individual questions should be combined into a second table that includes more RVF related questions (if asked, but not presented). This survey information seems under reported in this manuscript and could provide better context for the qualitative summaries, especially because the survey was described as phase 1 and the qualitative work as phase 2. Was the survey results used to create interview guides?

The results section can still describe the themes currently used by the author- RVF signs and symptoms, risk pathways, food preparation/consumption, ritual sacrifice, healing practices, incubation period, and gender roles. However, as opposed to editorializing in the results section and citing other manuscripts, the authors should highlight more of the findings and themes that emerged from the interviews and focus groups, as well as present more relevant quotes.

**Conclusions**

-Are the conclusions supported by the data presented?

-Are the limitations of analysis clearly described?

-Do the authors discuss how these data can be helpful to advance our understanding of the topic under study?

-Is public health relevance addressed?

Reviewer #1: (No Response)

Reviewer #2: The overall discussion and conclusions stated by the authors agree with the data presented. The authors state that the aim of documenting and quantifying the Somali pastoralists beliefs, knowledge and understanding of RVF was to better target prevention and intervention measures for RVF to these populations and specific nomadic populations. The authors also stated that understanding of these beliefs and knowledge would also be used to incorporate the pastoralists understanding of RVF into surveillance and help provide an “early warning” to potential RVF events, providing time to implement interventions and prevention. I believe the authors did make their point, I do not believe they provided strong enough evidence or provide much detail in how these findings would or how they might be used for any significant public health intervention or surveillance strategy. 

The potential limitations of the findings presented were not fully addressed. The survey, although well implemented and designed, did not seem to be detailed or comprehensive enough to provide the level of data needed to provide a higher level of evidence for how the knowledge and practices and risk factors identified for RVF in these populations and translate that into a detailed plan for public health action.

The authors do address how these data will be used to better understand the topic under study, but as stated previously, I do not think the data presented significantly advances the overall knowledge for RVF beyond what has been shown in previous studies. Although the population under study in this manuscript is a hard to reach and somewhat isolated population, the study did not show how this population differed that much from other similar populations in their risk or knowledge or behavior towards RVF. 

The public health relevance was address in that the authors would like to use this information and findings to help incorporate them into targeted health education and prevention measures. The authors did not detail out as well as they could the way this would be accomplished or how these messages would/should be implemented. In addition, the authors suggest incorporating these populations into “early warning” surveillance” they also did not detail out or show, in comparison to previously published work, how or the best method for doing this or practically how this might be accomplished given their remote locations and isolation which pose challenges to what the authors are proposing. In addition, the authors do not detail out how if they were successful in integrating these populations into routine RVF surveillance how that would impact or reduce the morbidity and mortality for RVF in this region or in Kenya at large.

Reviewer #3: The editorialization and citing of prior literature in the combined results/discussion section can be synthesized with the current conclusion section where each section is again briefly summarized. 

The manuscript currently does not have a limitations section. As described above, one of the limitations is the convenience sampling methodology and may not be representative of the entire Somali pastoralist community. 

Their overall recommendations involve strengthening public health efforts, however, more definitive examples are not given. Key informant interviews were conducted during phase 2 of their study, therefore it is implied that findings from the survey data were known prior to these interviews with health and veterinary officials and government administrators. If these interviews contained discussions on how to address these issues, then that should be presented here. For example, how can early warning surveillance systems incorporate pastoralists? By providing training? By providing cell phones and a reporting mechanism? By training local veterinarians to report abnormal symptoms/abortions in cattle to the local government? Likewise, the authors also recommend public health campaigns to address socio-cultural practices, but do not elaborate what this would involve. What would help with high-risk exposures- use of PPE? Working with community and religious leaders? Overall, the barriers identified from their KAP survey have previously been reported in regards to RVF in prior publications, but their qualitative work likely has potential answers on how to intervene on these identified barriers. 

The end of the conclusion ends with a summary of the gender findings described in the previous results section. The manuscript would be stronger if it ended with a more encompassing summary statement.

**Editorial and Data Presentation Modifications?**

Reviewer #1: (No Response)

Reviewer #2: - Figures are clear and revised figures are clear. 

- Addition of a detailed map would be beneficial

Reviewer #3: - The authors state that humans usually get RVF through bites from infected mosquitos, without a reference. This is actually debatable and in prior RVF outbreaks exposure to animals through slaughtering, veterinary care, or other contact with bodily fluids of infected animals has been implicated over mosquito to human transmission

-The 2018 RVF outbreak in Kenya's Wajir and Marsabit counties was not mentioned in the paper. There was reported increase in animal death and abortions in Garissa county. Was this study done after or before this recent outbreak? The study period was not defined.

https://www.who.int/csr/don/18-june-2018-rift-valley-fever-kenya/en/

**Summary and General Comments**

Reviewer #1: Please see attached "Reviewer Comments".

Reviewer #2: The authors of “Indigenous knowledge of Rift Valley Fever among Somali nomadic pastoralists and its

implications on public health delivery approaches in Ijara Division, North Eastern Kenya” describe the knowledge and practices that show this population may be aware of the risk factors for RVF but may also not be practicing safe and preventative measures which also keeps them at high risk for RVF. The authors preformed a series of interviews and analysis showing that this population does have a high knowledge of RVF. They also show how some sub populations of these populations may be at higher risk than others and should have more targeted messaging that may differ from more general health prevention messaging. 

My concerns are that although the study was nicely conducted and results were clearly described, the results and outcomes were not too different than what we have seen from other studies. The authors would have benefited by highlighting that although the risk factors and their knowledge of RVF was similar to other nomadic populations, the unique characteristics of this population could greatly benefit from specific targeted health messaging and they could provide valuable information to public health authorities for helping to prevent the next large epizootic of RVF in NE Kenya. The authors do state this in general, they do not go into any great detail which would make the results of this manuscript be unique and stand out in its utilization of the already common knowledge conclusions to benefit public health. It would be beneficial of the public health benefits from the results of this study were more detailed and descriptive.

Reviewer #3: This manuscript by Dr. Muga, et al assesses RVF knowledge among Somali nomadic pastoralists in North Eastern Kenya. The authors administered a survey to 204 herd owners; conducted in depth interviews with health, veterinary, and governmental officials; conducted 5 narrative interviews with individuals impacted by RVF; and held 4 FGD with herd owners. Overall, the scope of the project was quite impressive and the qualitative component makes it unique from the several prior RVF KAP survey studies that have been previously published and would be a welcomed addition to the RVF literature. However, the manuscript would be stronger if it reported several key elements, including participant demographics and more survey data. The qualitative methods also need to be better described and elements from the interviews with key stakeholders should be emphasized and incorporated into public health recommendations.

PLOS authors have the option to publish the peer review history of their article (what does this mean?). If published, this will include your full peer review and any attached files.

Reviewer #1: No

Reviewer #2: No

Reviewer #3: No
---

## [Decision Letter · Decision Letter 1]

10 Aug 2020

Dear Dr Muga,

Thank you very much for submitting your manuscript "Indigenous knowledge of Rift Valley Fever among Somali nomadic pastoralists and its implications on public health delivery approaches in Ijara Sub-County, North Eastern Kenya" for consideration at PLOS Neglected Tropical Diseases. As with all papers reviewed by the journal, your manuscript was reviewed by members of the editorial board and by several independent reviewers. In light of the reviews (below this email), we would like to invite the resubmission of a significantly-revised version that takes into account the reviewers' comments. 

We cannot make any decision about publication until we have seen the revised manuscript and your response to the reviewers' comments. Your revised manuscript is also likely to be sent to reviewers for further evaluation.

Sincerely,

Benjamin Althouse

Deputy Editor

Scott Weaver

Deputy Editor

Reviewer's Responses to Questions

**Key Review Criteria Required for Acceptance?**

**Methods**

-Are the objectives of the study clearly articulated with a clear testable hypothesis stated?

-Is the study design appropriate to address the stated objectives?

-Is the population clearly described and appropriate for the hypothesis being tested?

-Is the sample size sufficient to ensure adequate power to address the hypothesis being tested?

-Were correct statistical analysis used to support conclusions?

-Are there concerns about ethical or regulatory requirements being met?

Reviewer #1: (No Response)

Reviewer #3: Overall, comments were adequately addressed in the methods section:

-The addition of demographic data was helpful to better understand the study population

-Inclusion and exclusion criteria have been defined

-Recruitment methods are more clearly defined

-Sample size calculation is now more clear

Further comments to address:

-“Secondary data review” section is more vague now. The current version eliminated what sources were reviewed (publications, books, newspaper reports, etc..) and just states that it was “undertaken.” In general, a literature review is assumed. Consider deleting this section. 

-Regarding interviews, were there multiple reviewers/coders? Who “cleaned and coded into running themes”? If one person, was it validated by another reviewer?

-It is unnecessary to state that data was stored on removable disks and hardcopy files. This is assumed and is included in a protocol or IRB submission, not a publication. It also follows an unrelated sentence. Consider removing.

**Results**

-Does the analysis presented match the analysis plan?

-Are the results clearly and completely presented?

-Are the figures (Tables, Images) of sufficient quality for clarity?

Reviewer #1: (No Response)

Reviewer #3: Comments to address

-Table 1. This table can be condensed. Would reformat to remove “total” from every variable. It would be more concise if there was one column for variable name and a second column with n(%). Also, income unit is not defined.

-Figures 2,3 and 4 should have titles that clearly outline what is being displayed. For example, figure 2 should specify that these are referring to symptoms in animals.

-Consider consolidating survey data that is sporadically cited throughout this section into a table. It would be useful to the reader to be able to review all of the KAP variables more easily and in one place. Perhaps in a supplement if not in the text?

**Conclusions**

-Are the conclusions supported by the data presented?

-Are the limitations of analysis clearly described?

-Do the authors discuss how these data can be helpful to advance our understanding of the topic under study?

-Is public health relevance addressed?

Reviewer #1: (No Response)

Reviewer #3: Overall, the specific recommendations for public health change make the manuscript much stronger and relevant. However, if true, recommend more clearly stating that these ideas came from the qualitative work (and not the authors’ ideas). For example, did the pastoralists and public health officials recommend using mobile phones? If so, then there would inherently be more acceptability and feasibility for an intervention such as this. 

The strengths and limitation section is currently within the methods section, but would be more appropriate in the conclusions section. Elements from the strengths portion would be a good way to end the manuscript, as opposed to how it ends now with a single specific recommendation regarding messaging. The limitations section needs to be flushed out more. For example, due to convenience sampling, the study population may not represent the true population of Ijara sub-county.

**Editorial and Data Presentation Modifications?**

Reviewer #1: (No Response)

Reviewer #3: The abstract does not specify how many people were surveyed or participated in interviews, and focus group discussions.

Even though the current study occurred prior to the 2018 outbreak, it should be listed in the introduction sentence that sites the 1997 and 2006 outbreaks. It further strengthens the 10-year interval cited by the authors.

There are grammatical errors throughout the manuscript that need to be revised.

**Summary and General Comments**

Reviewer #1: (No Response)

Reviewer #3: Overall, this current version is much improved. In addition to the qualitative and survey data, the recommendations regarding improving early warning systems and community messaging are an important addition to the literature on RVF.

PLOS authors have the option to publish the peer review history of their article (what does this mean?). If published, this will include your full peer review and any attached files.

Reviewer #1: Yes: Kennedy Wanjala, PhD

Reviewer #3: No
---

## [Decision Letter · Decision Letter 2]

15 Dec 2020

Dear Dr Muga,

Thank you very much for submitting your manuscript "Indigenous knowledge of Rift Valley Fever among Somali nomadic pastoralists and its implications on public health delivery approaches in Ijara sub-County, North Eastern Kenya" for consideration at PLOS Neglected Tropical Diseases. As with all papers reviewed by the journal, your manuscript was reviewed by members of the editorial board and by several independent reviewers. The reviewers appreciated the attention to an important topic. Based on the reviews, we are likely to accept this manuscript for publication, providing that you modify the manuscript according to the review recommendations. 

Please address the remaining reviewer comments and make sure the manuscript is free from grammatical errors. There are currently quite a few.

Sincerely,

Benjamin Muir Althouse

Deputy Editor

Scott Weaver

Deputy Editor

Please address the remaining reviewer comments and make sure the manuscript is free from grammatical errors. There are currently quite a few.

Reviewer's Responses to Questions

**Key Review Criteria Required for Acceptance?**

**Methods**

-Are the objectives of the study clearly articulated with a clear testable hypothesis stated?

-Is the study design appropriate to address the stated objectives?

-Is the population clearly described and appropriate for the hypothesis being tested?

-Is the sample size sufficient to ensure adequate power to address the hypothesis being tested?

-Were correct statistical analysis used to support conclusions?

-Are there concerns about ethical or regulatory requirements being met?

Reviewer #3: The authors adequately addressed the prior recommendations. The secondary data review and interview transcription, coding, and validation methods are all more clearly defined.

**Results**

-Does the analysis presented match the analysis plan?

-Are the results clearly and completely presented?

-Are the figures (Tables, Images) of sufficient quality for clarity?

Reviewer #3: Table 1 is more concise now that a separate row for total has been removed for each variable. However, it is best practive for the the table to still specify either the denominator (N=204) in the header/title or the numerator should be listed next to each percentage (e.g., Sex Male 161(79.4%). 

The figures 3 and 4 are now labeled and the presented content is more evident. The authors should also consider not rounding to the 100th decimal point- whole number or 10th is sufficient for the percentages presented. As in table 1, neither the numerator or denominator are presented. The denominator (N=204) could be presented in the title, footnote, or consider presenting the numerator (%) above each bar.

**Conclusions**

-Are the conclusions supported by the data presented?

-Are the limitations of analysis clearly described?

-Do the authors discuss how these data can be helpful to advance our understanding of the topic under study?

-Is public health relevance addressed?

Reviewer #3: The expansion on mobile phone uptake and implications for notification is better emphasized. Likewise, the conclusion section provides a better summary and interpretation of the findings now.

**Editorial and Data Presentation Modifications?**

Reviewer #3: (No Response)

**Summary and General Comments**

Reviewer #3: Overall, this revised version is significantly improved from prior submissions. The authors have done a sufficient job utilizing the comments from all 3 reviewers to improve this manuscript. The data and findings presented in this report are an important contribution to the literature on RVF. I only have several several minor recommendations to further clarify the data presented.

PLOS authors have the option to publish the peer review history of their article (what does this mean?). If published, this will include your full peer review and any attached files.

Reviewer #3: No
---

## [Editor Report · Decision Letter 3]

21 Jan 2021

Dear Dr Muga,

We are pleased to inform you that your manuscript 'Indigenous knowledge of Rift Valley Fever among Somali nomadic pastoralists and its implications on public health delivery approaches in Ijara sub-County, North Eastern Kenya' has been provisionally accepted for publication in PLOS Neglected Tropical Diseases.

Best regards,

Benjamin Muir Althouse

Deputy Editor

Scott Weaver

Deputy Editor

---

## [Editor Report · Acceptance letter]

16 Feb 2021

Dear Dr Muga,

We are delighted to inform you that your manuscript, "Indigenous knowledge of Rift Valley Fever among Somali nomadic pastoralists and its implications on public health delivery approaches in Ijara sub-County, North Eastern Kenya," has been formally accepted for publication in PLOS Neglected Tropical Diseases.

Best regards,

Shaden Kamhawi

co-Editor-in-Chief

Paul Brindley

co-Editor-in-Chief
